# The Prediction of Gestational Hypertension, Preeclampsia and Fetal Growth Restriction via the First Trimester Screening of Plasma Exosomal C19MC microRNAs

**DOI:** 10.3390/ijms20122972

**Published:** 2019-06-18

**Authors:** Ilona Hromadnikova, Lenka Dvorakova, Katerina Kotlabova, Ladislav Krofta

**Affiliations:** 1Department of Molecular Biology and Cell Pathology, Third Faculty of Medicine, Charles University, 10000 Prague, Czech Republic; lenka.dvorakova@lf3.cuni.cz (L.D.); katerina.kotlabova@lf3.cuni.cz (K.K.); 2Institute for the Care of the Mother and Child, Third Faculty of Medicine, Charles University, 14700 Prague, Czech Republic; ladislav.krofta@upmd.eu

**Keywords:** C19MC microRNA, expression, exosomes, fetal growth restriction, gestational hypertension, plasma, prediction, preeclampsia, pregnancy-related complications, screening

## Abstract

The aim of the study was to verify if quantification of placental specific C19MC microRNAs in plasma exosomes would be able to differentiate during the early stages of gestation between patients subsequently developing pregnancy-related complications and women with the normal course of gestation and if this differentiation would lead to the improvement of the diagnostical potential. The retrospective study on singleton Caucasian pregnancies was performed within 6/2011-2/2019. The case control study, nested in a cohort, involved women that later developed GH (*n* = 57), PE (*n* = 43), FGR (*n* = 63), and 102 controls. Maternal plasma exosome profiling was performed with the selection of C19MC microRNAs with diagnostical potential only (miR-516b-5p, miR-517-5p, miR-518b, miR-520a-5p, miR-520h, and miR-525-5p) using real-time RT-PCR. The down-regulation of miR-517-5p, miR-520a-5p, and miR-525-5p was observed in patients with later occurrence of GH and PE. Maternal plasma exosomal profiling of selected C19MC microRNAs also revealed a novel down-regulated biomarker during the first trimester of gestation (miR-520a-5p) for women destinated to develop FGR. First trimester circulating plasma exosomes possess the identical C19MC microRNA expression profile as placental tissues derived from patients with GH, PE and FGR after labor. The predictive accuracy of first trimester C19MC microRNA screening (miR-517-5p, miR-520a-5p, and miR-525-5p) for the diagnosis of GH and PE was significantly higher in the case of expression profiling of maternal plasma exosomes compared to expression profiling of the whole maternal plasma samples.

## 1. Introduction

Most previous studies performed C19MC microRNA profiling analyses on whole maternal plasma or serum samples with the aim to diagnose or predict the later occurrence of pregnancy-related complications [1,2,3,4,5,6,7,8,9,10]. Nevertheless, nowadays with regard to the perspectives of potential usage of exosomes as therapeutics in placental-mediated disorders [11,12,13,14], it is crucial to characterize first the inner content of placental derived exosomes, and next to describe their impact on the modulation of maternal immune system and metabolism through the mediation of distant cell–cell communication [11]. 

Exosomes are small size vesicles (30–150 nm) of the endosomal origin released to the extracellular space by most cells including trophoblast cells that mediate cell–cell communication through signaling molecules (proteins, lipids, RNA, and DNA) released after the exocytosis fusion of multi-vesicular body with the cell membrane of the target cell [11,15,16,17,18,19,20,21,22,23].

C19MC microRNAs represent unique placental specific biomarkers to be tested in plasma/serum exosomes during gestation, since only paternally inherited alleles are expressed in the placenta due to genomic imprinting [24]. Nevertheless, since some microRNAs from C19MC microRNA cluster were demonstrated to be also expressed in the testis, embryonic stem cells, and specific tumors [25,26,27,28], we previously selected, from the C19MC microRNA cluster, only those microRNAs (miR-516-5p, miR-517-5p, miR-518b, miR-520a-5p, miR-520h, and miR-525-5p) that were exclusively or abundantly expressed in the placenta, showed minimal expression in other tissues and maximum diagnostical potential (100% detection rate in maternal plasma samples throughout gestation, from early stages to term pregnancy) [3,29,30].

To date, little data on first trimester exosome microRNA profiling is available in women with subsequent development of pregnancy-related complications such as gestational hypertension (GH), preeclampsia (PE), and/or fetal growth restriction (FGR) [22,31].

This study is a follow-up of our previous studies dedicated to first trimester screening of circulating C19MC microRNAs in whole maternal plasma and its potential to predict subsequent onset of gestational hypertension, preeclampsia, and/or FGR [6,9]. The aims of the current study are to explore A) if quantification of placental specific C19MC microRNAs in plasma exosomes would be able to differentiate during the early stages of gestation between patients subsequently developing pregnancy-related complications and women with normal course of gestation and B) if this differentiation would lead to the improvement of their diagnostical potential (better detection rate).

## 2. Results

Gene expression of C19MC microRNAs in plasma exosomes was retrospectively compared between women with normal and complicated course of gestation (GH, PE, and FGR) within 10 to 13 weeks. Just the results that reached a statistical significance or displayed a trend toward aberrant levels of circulating C19MC microRNAs in complicated cases are presented below.

### 2.1. Plasma Exosomal miR-517-5p, miR-520a-5p and miR-525-5p are Down-Regulated during the First Trimester of Gestation in Women Affected with GH and PE

The expression profile of miR-517-5p, miR-520a-5p, and miR-525-5p differed significantly or showed a trend toward statistical significance between the groups of women with later onset of GH or PE and the controls. Decreased levels of miR-517-5p, miR-520a-5p, and miR-525-5p were detected during the first trimester of gestation in circulating plasma exosomes in women that later developed either GH or PE (Figure 1).

### 2.2. The High Accuracy of First Trimester C19MC MicroRNA Expression Profiling in Maternal Plasma Exosomes to Identify Women at a Risk of Later Development of GH or PE

The screening of individual C19MC microRNA biomarkers in plasma exosomes directed to the prediction of subsequent onset of GH reached a very high accuracy (miR-517-5p: AUC 0.812, *p* < 0.001; miR-520a-5p: AUC 0.806, *p* < 0.001; and miR-525-5p: AUC 0.802, *p* < 0.001). The predictive performance of miR-517-5p, miR-520a-5p, and miR-525-5p reached 48.21%, 57.14%, and 57.14% at 10.0% false positive rate (FPR) (Figure 2).

The combination of miR-520a-5p and miR-525-5p (AUC 0.808, *p* < 0.001) had an advantage over using the miR-517-5p biomarker only (AUC 0.812, *p* < 0.001), since it was able to predict a significantly higher number of women that later developed GH (66.07% sensitivity at 10.0% FPR vs. 48.21% sensitivity at 10.0% FPR) (Figure 3).

Similarly, the ROC curve analyses, revealed significantly lower levels of miR-517-5p (AUC 0.634, *p* = 0.022, 27.91% sensitivity at 10.0% FPR), miR-520a-5p (AUC 0.699, *p* < 0.001, 41.86% sensitivity at 10.0% FPR), and miR-525-5p (AUC 0.698, *p* < 0.001, 39.53% sensitivity at 10.0% FPR) in a substantial proportion of mothers during the first trimester of gestation that subsequently developed PE (Figure 4).

The combined screening of miR-517-5p, miR-520a-5p, and miR-525-5p was superior over using individual C19MC microRNA biomarkers or their dual combinations, since it was able to predict the highest number of women with the subsequent onset of PE (AUC 0.719, *p* < 0.001, 44.19% sensitivity at 10.0% FPR) (Figure 5).

### 2.3. MiR-520a-5p Represents a Novel Maternal Plasma Exosome C19MC MicroRNA Biomarker for Prediction of Later Onset of FGR

The Kruskal-Wallis test indicated the down-regulation of miR-520a-5p in circulating plasma exosomes during the first trimester of gestation in women carrying subsequently growth-restricted fetuses (Figure 6). 

The predictive performance of miR-520a-5p reaches 22.22% sensitivity at 10.0% FPR (AUC 0.611, *p* = 0.037) (Figure 7). 

## 3. Discussion

To our knowledge, no data on C19MC microRNA profiling in maternal plasma exosomes during the first trimester of gestation was reported.

We found out that first trimester circulating plasma exosomes possess the identical C19MC microRNA expression profile as placental tissues derived from patients with GH, PE, and FGR after labor [32]. 

In placental tissues, down-regulation of 4/15 tested C19MC microRNAs (miR-517-5p, miR-519d, miR-520a-5p, and miR-525-5p) was previously observed in GH patients [32]. After the investigation of maternal plasma exosome C19MC microRNA expression profile during the first trimester of gestation with the selection of C19MC microRNAs with diagnostical potential only (miR-516b-5p, miR-517-5p, miR-518b, miR-520a-5p, miR-520h, and miR-525-5p) [29], we observed the down-regulation of miR-517-5p, miR-520a-5p, and miR-525-5p in patients with later occurrence of GH, which is in compliance with the findings in placental tissues derived from GH patients during the delivery [32]. 

Similarly, in PE patients, down-regulation of 11/15 tested C19MC microRNAs (miR-515-5p, miR-517-5p, miR-518b, miR-518f-5p, miR-519a, miR-519d, miR-520a-5p, miR-520h, miR-524-5p, miR-525-5p, and miR-526a) was previously demonstrated [32]. Parallel, after the testing of C19MC microRNAs with diagnostical potential only (miR-516b-5p, miR-517-5p, miR-518b, miR-520a-5p, miR-520h, and miR-525-5p) [29] in maternal plasma exosomes during the first trimester of gestation, we identified decreased levels of miR-517-5p, miR-520a-5p, and miR-525-5p in patients that subsequently developed PE, which corresponds to a considerable extent to the expression profiles observed in PE affected placental tissues [32]. 

Nevertheless, these data are opposed to the results of our previous studies [6,9], indicating first trimester upregulation of circulating C19MC microRNAs in maternal plasma with predictive accuracy of subsequent development of gestational hypertension (miR-516-5p, miR-517-5p, miR-518b, miR-520a-5p, and miR-520h) or preeclampsia (miR-517-5p). Other investigators also observed increased levels of some C19MC microRNAs (miR-520) in sera from 12 to 14 weeks of gestation in women, who later developed severe preeclampsia [7]. We believe that dissimilar expression profiles of C19MC microRNAs between maternal plasma and maternal plasma exosomes can be influenced by compilations stemming from several factors. At the very least, an expression of particular C19MC microRNA in maternal plasma is represented by the total sum of expression of this particular C19MC microRNA in individual cells located in different areas of placenta, which currently undergo apoptosis, release placental debris into the maternal circulation, and actively secrete exosomes mediating intercellular communication [9]. 

However, the predictive accuracy of first trimester C19MC microRNA screening for the diagnosis of gestational hypertension is significantly higher in case of expression profiling of miR-517-5p (AUC 0.812, *p* < 0.001 vs. AUC 0.752, *p* = 0.002) and miR-520a-5p: (AUC 0.806, *p* < 0.001 vs. AUC 0.688, *p* = 0.031) in maternal plasma exosomes compared to expression profiling of whole maternal plasma samples [6]. 

In case of preeclampsia, the predictive accuracy of miR-517-5p is nearly identical for maternal plasma exosomes and whole maternal plasma samples (AUC 0.634, *p* = 0.022 vs. AUC 0.700, *p* = 0.045) [9]. But, maternal plasma exosome C19MC microRNA profiling significantly improved the predictive accuracy of miR-520a-5p (AUC 0.699, *p* < 0.001 vs. AUC 0.495, *p* = 0.951) and miR-525-5p (AUC 0.698, *p* < 0.001 vs. AUC 0.475, *p* = 0.755) for preeclampsia [9]. In addition, the best predictive performance for preeclampsia was achieved when maternal plasma exosome combined profiling of miR-517-5p, miR-520a-5p, and miR-525-5p (AUC 0.719, *p* < 0.001) was performed. 

In case of FGR, placental tissues showed down-regulation of 6/15 tested C19MC microRNAs (miR-517-5p, miR-518f-5p, miR-519a, miR-519d, miR-520a-5p, and miR-525-5p) [32]. Maternal plasma exosomal profiling of selected C19MC microRNAs [29] revealed a novel down-regulated biomarker during the first trimester of gestation (miR-520a-5p) for women destinated to develop FGR, which was not identified when whole maternal plasma analysis was performed [9].

This study confirmed the former hypothesis, that the exosomes released to the systemic circulation represent unique non-invasive source of signalling molecules, including microRNAs, whose aberrant expression profile reflects expression profile of the parent cells (in this particular event the trophoblast cells) [22,23]. These observations support the idea that placenta-derived exosomes may be utilized as a part of first trimester screening to identify a significant proportion of women at a risk of later development of pregnancy-related complications such as gestational hypertension, preeclampsia, and FGR [22]. The only weakness of this approach is that the screening of C19MC microRNAs in plasma exosomes is not able to differentiate during the first trimester of gestation between the women that later develop GH and those ones that later develop PE, since the down-regulation of the same biomarkers (miR-517-5p, miR-520a-5p, and miR-525-5p) is present from early gestation. Nevertheless, through the mediation of this approach, novel microRNA biomarkers may be identified at some time in the future, which would be able to differentiate between women at a risk of GH and at risk of PE, to enable the primary prevention of preeclampsia via the early administration of low-dose aspirin [33,34,35]. 

Nevertheless, recent findings confirmed that even in women at a high risk of pregnancies with small-for-gestational-age foetuses the administration of aspirin at a dose of ≥100 mg starting at or before 16 weeks of gestation is recommended [36,37,38]. Therefore, miR-520a-5p may be a novel promising placental specific biomarker for FGR with a potential of early stratification of high-risk pregnancies, which may benefit from primary prevention strategies as well.

## 4. Materials and Methods 

### 4.1. Patients Cohort

The study had a retrospective design, it was performed in 6/2011–2/2019. The study cohort involved singleton pregnancies of Caucasian descent only. Of 4356 women undergoing first trimester screening at 10–13 weeks of gestation, 3092 women were finally followed-up and delivered in the Institute for the Care of Mother and Child, Prague, Czech Republic, 1189 women were followed-up and delivered in another health care provider, and in 75 women gestation was terminated for fetal anomaly or missed abortion appeared. 

The case control study nested in a cohort involved women that later developed relevant pregnancy-related complications (57 GH, 43 PE, and 63 FGR) [39,40,41,42]. Finally, 13 of 43 PE patients developed mild PE, 30 of 43 PE patients suffered from severe PE, 10 of 43 PE patients were diagnosed with early PE (before 34 week of gestation) and 33 of 43 PE patients delivered after 34 week of gestation (late PE) [39,40,41]. Superimposed preeclampsia occurred in 5 out of 43 cases [39,40,41]. 

Of 63 pregnancies complicated with FGR, 4 foetuses were delivered before 32 week of gestation (early FGR), other 59 cases were diagnosed with late form of FGR (diagnosed after 32 week of gestation) [42]. 

Oligohydramnios or anhydramnios were found in 1 PE case and 16 FGR-affected foetuses. 

Aberrant index of pulsatility (PI) was detected in arteria umbilicalis (above 95th percentile, 3 PE cases, 23 FGR cases), arteria cerebri media (below 5th percentile, 3 PE cases, 14 FGR cases), arteria uterine (above 95th percentile, 7 PE case, 4 FGR cases), and Ductus venosus (>1, 3 FGR cases) [43,44]. The aberrant cerebro-placental ratio (CPR), below 5th percentile, was detected in 7 PE cases and 47 FGR cases [45,46,47,48]. Absent and/or zero diastolic flow in the arteria umbilicalis was present in 3 FGR cases [49,50]. An absence of flow during atrial contraction in Ductus venosus was detected in 1 FGR case [51,52]. The presence of unilateral or bilateral diastolic notch in the uterine artery was observed in 7 PE cases and 5 FGR cases [53,54]. 

The control group, normal pregnancies without complications delivering full term, healthy infants after 37 weeks of gestation weighting >2500 g, was selected on the basis of equal gestational age, equal age of women at the time of sampling and equal plasma sample storage times. The control group was separated into two subgroups and involved 102 cases altogether (the group 1 consisted of 50 cases, and the group 2 of 52 cases). 

The clinical characteristics of the controls and complicated pregnancies are outlined in Table 1.

Written informed consent was provided for all participants included in the study. The study was approved by the Ethics Committee of the Third Faculty of Medicine, Prague, Czech Republic (Implication of placenta-specific microRNAs in maternal circulation for diagnosis and prediction of placental insufficiency, date of approval: 7.4.2011).

### 4.2. Processing of Samples

Two millilitres of incoagulable peripheral blood (EDTA tubes) were centrifuged twice immediately after collection at 1200 rcf (4600 rpm) for 10 min at room temperature. Plasma samples were then stored frozen at −80 °C until further processing.

### 4.3. Isolation and Purification of Exosomes from Maternal Plasma Samples

Exosomes were isolated from 0.6 mL of maternal plasma samples using miRCURY^TM^ Exosome Isolation Kit-Serum and plasma (Exiqon, Woburn, MA, USA, no: 300101) according to the manufacturer´s instructions. After the exosome isolation and purification, RNA was isolated immediately from 200 µL supernatant using miRCURY^TM^ RNA Isolation Kit-Biofluids (Exiqon, Woburn, MA, USA, no: 300112) according to manufacturer’s instructions. After 3 min incubation of 200 µL supernatant with 60 µL Lysis Solution buffer, 1 µL RNA spike-in (1 nM cell-miR-39, synthetic C. elegans microRNA, Qiagen, Hilden, Germany, no: MSY0000010) and 20 µL Protein Precipitation Solution buffer were added into the mixture. In order to maximize yield of exosomal RNA, total elution volume of 100 µL was used (50 µL in 2 steps eluting with half of the recommended total volume each). DNA contamination of RNA was removed by the 30 min treatment of eluted RNA with 5 µL DNase I (Thermo Fisher Scientific, Waltham, MA, USA, no: EN0521) at 37 °C.

The quality of the isolated exosomes was not checked using flow cytometry, electron microscopy, or other techniques, since we were not interested in exosomal subpopulations present in analysed samples or in performance of exosomal functional studies. The protocols of miRCURY™ Exosome Isolation Kit—Serum and plasma (Exiqon, Woburn, MA, USA, no: 300101) are validated to allow subsequent microRNA isolation using the miRCURY™ RNA Isolation Kit - Biofluids (Exiqon, Woburn, MA, USA, no: 300112) and improve the quality of the obtained microRNA signature.

### 4.4. Reverse Transcription Reaction

The analyzed C19MC microRNAs and cell-miR-39 were reverse transcribed into complementary DNA using TaqMan^TM^ MicroRNA Assays (Thermo Fisher Scientific, Waltham, MA, USA, miR-516b-5p no: 001281, miR-517-5p no: 001113, miR-518b no: 001156, miR-520a-5p no: 001168, miR-520h no: 001170, miR-525-5p no: 001174, and cell-miR-39 no: 000200), and TaqMan MicroRNA Reverse Transcription Kit (Thermo Fisher Scientific, Waltham, MA, USA, no: 4366597). Reverse transcription reaction was performed in a total reaction volume of 32 µL in case of C19MC microRNAs and in a total reaction volume of 10 µL in case of cell-miR-39 on a 7500 Real-Time PCR system (Thermo Fisher Scientific, Waltham, MA, USA) under predefined thermal cycling parameters: 30 min at 16 °C, 30 min at 42 °C, 5 min at 85 °C, and then held at 4 °C [3,5,6,9,32].

### 4.5. Quantification of Plasma Exosomal C19MC microRNAs by Real-Time PCR

15 µL of cDNA corresponding to C19MC microRNAs and 4.4 µL of cDNA corresponding to cell-miR-39 were mixed with components of TaqMan MicroRNA Assays (Thermo Fisher Scientific, Waltham, MA, USA, miR-516b-5p no: 001281, miR-517-5p no: 001113, miR-518b no: 001156, miR-520a-5p no: 001168, miR-520h no: 001170, miR-525-5p no: 001174, and cell-miR-39 no: 000200), and the ingredients of the TaqMan Universal PCR Master Mix (Thermo Fisher Scientific, Waltham, MA, USA, no: 4318157). The analysis was performed using a 7500 Real-Time PCR System under the conditions described in the TaqMan guidelines in a total reaction volume of 35 µL. All PCRs were performed in duplicates. A sample displaying the amplification signal before the 40th threshold cycle (Ct) was considered positive. 

The expression of particular C19MC microRNA in maternal plasma exosomes was determined using the comparative Ct method [55] relative to the expression in the reference sample. RNA isolated from the pool of randomly selected maternal plasma samples derived from women at the first trimester with normal course of gestation was used as a reference sample for relative quantification. Two reference samples were used throughout the study (reference 1: the pool of 5 maternal plasma samples, reference 2: the pool of 8 maternal plasma samples). 

Real-time PCR data were normalized to synthetic C. elegans microRNA (cell-miR-39, Qiagen, Hilden, Germany, no: MSY0000010) showing no sequence homology to any human microRNA: 2^−ΔΔCt^ = [(Ct particular C19MC microRNA—Ct cel-miR-39) tested sample—(Ct particular C19MC microRNA—Ct cel-miR-39) reference sample] [6,9].

### 4.6. Statistical Analysis

Normality of the data was assessed using Shapiro-Wilk test, which indicated that our experimental data did not follow a normal distribution (Appendix A). Therefore, C19MC microRNA levels were primarily compared between groups using non-parametric test (the Kruskal-Wallis one-way analysis of variance with post-hoc test for the comparison among multiple groups). The significance level was established at a *p*-value of *p* < 0.05. 

Receivers operating characteristic (ROC) curves were constructed to calculate the area under the curve (AUC) and the best cut-off point for particular C19MC microRNA was used in order to calculate the respective sensitivity at 90.0% specificity, respectively (MedCalc Software bvba, Ostend, Belgium). For every possible threshold or cut-off value, the MedCalc program reports the sensitivity, specificity, likelihood ratio positive (LR+), and likelihood ratio negative (LR-).

To select the optimal combinations of C19MC microRNA biomarkers logistic regression was applied (MedCalc Software bvba, Ostend, Belgium). To perform a full ROC curve analysis the predicted probabilities were first saved and next used as a new variable in ROC curve analysis. The dependent variable used in logistic regression acted as the classification variable in the ROC curve analysis dialog box.

Box plots encompassing the median (the Kruskal-Wallis test) of gene expression values for particular C19MC microRNAs were generated using Statistica software (version 9.0; StatSoft, Inc., Tulsa, OK, USA). The upper and lower limits of the boxes represent the 75th and 25th percentiles (the Kruskal-Wallis test), respectively. The upper and lower whiskers indicate the maximum and minimum values that are no more than 1.5 times the span of the interquartile range (range of the values between the 25th and the 75th percentiles) (the Kruskal-Wallis test). Outliers are marked by circles, and extremes by asterisks.

The presentation of no statistically significant results is provided in Appendix A.

## 5. Conclusions

The down-regulation of miR-517-5p, miR-520a-5p, and miR-525-5p was observed in patients with later occurrence of GH and PE. Maternal plasma exosomal profiling of selected C19MC microRNAs also revealed a novel down-regulated biomarker during the first trimester of gestation (miR-520a-5p) for women destinated to develop FGR. First trimester circulating plasma exosomes possess the identical C19MC microRNA expression profile as placental tissues derived from patients with GH, PE and FGR during labour. The predictive accuracy of first trimester C19MC microRNA screening (miR-517-5p, miR-520a-5p, and miR-525-5p) for the diagnosis of GH and PE was significantly higher in case of expression profiling of maternal plasma exosomes compared to expression profiling of whole maternal plasma samples. Consecutive large-scale studies are needed to verify the findings resulting from this pilot study. Nevertheless, the performance of that kind of studies will be highly demanding, since ten thousand of the first trimester plasma samples have to be collected to get sufficient amount of cases who will subsequently develop pregnancy-related complications such as GH, PE, or FGR. For the purpose of this study we collected plasma samples from 4356 women to acquire 163 samples from women that later developed relevant pregnancy-related complications (57 GH, 43 PE, and 63 FGR).

## Figures and Tables

**Figure 1 ijms-20-02972-f001:**
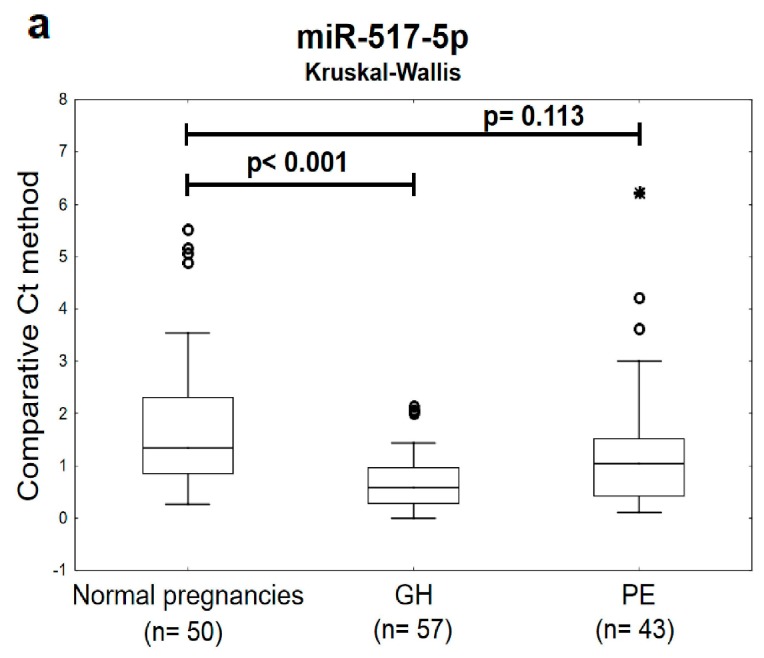
Down-regulation of miR-517-5p, miR-520a-5p, and miR-525-5p in plasma exosomes during the first trimester of gestation in women with later onset of GH or PE.(**a**–**c**) Decreased levels of miR-517-5p, miR-520a-5p, and miR-525-5p were observed in circulating plasma exosomes within 10 to 13 weeks of gestation in women affected with GH or PE when the comparison to the controls was performed using non-parametric statistical test (the Kruskal-Wallis test). GH: gestational hypertension; PE: preeclampsia. Outliers are marked by circles (⸰), and extremes by asterisks (*).

**Figure 2 ijms-20-02972-f002:**
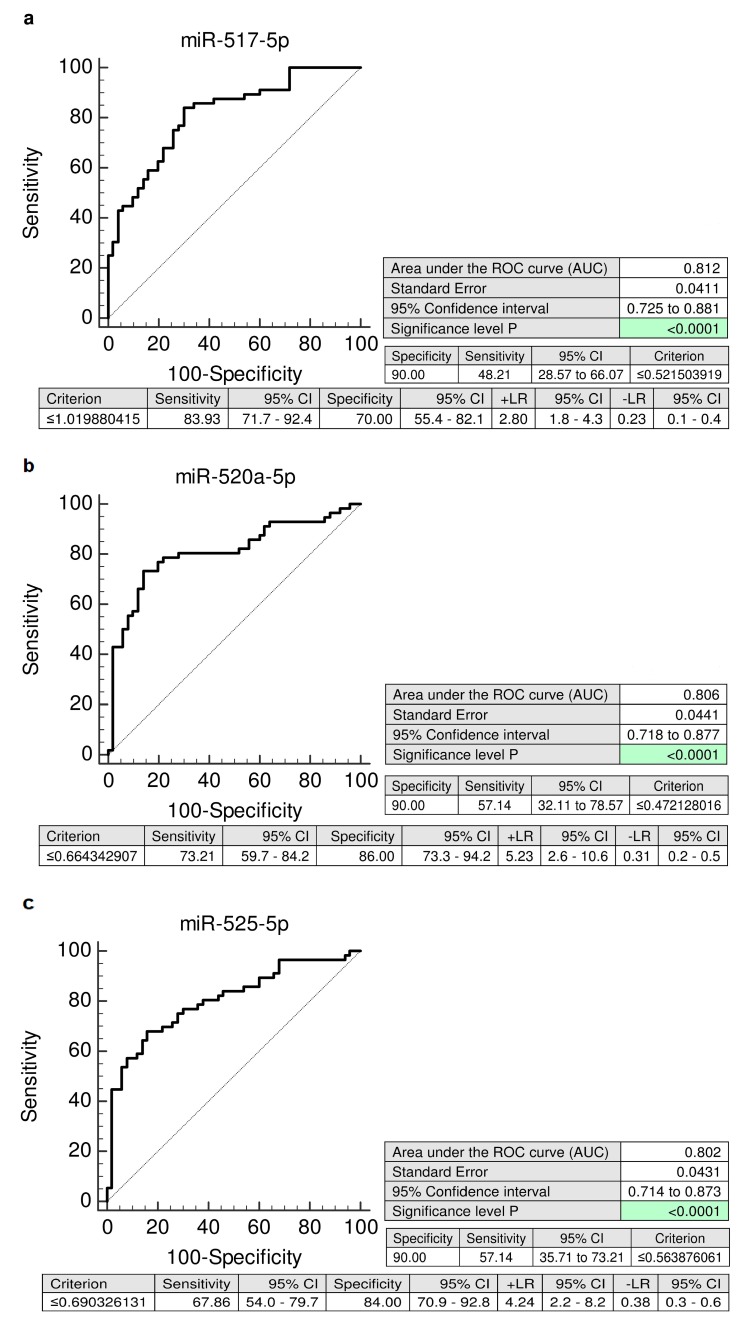
ROC curves—individual C19MC microRNA biomarkers—evaluation of the potential of the first trimester C19MC microRNA screening in plasma exosomes to predict later onset of GH. Decreased levels of miR-517-5p, miR-520a-5p, and miR-525-5p were detected in women destinated to develop GH when the comparison to the controls was performed both (**a**–**c**). GH: gestational hypertension.

**Figure 3 ijms-20-02972-f003:**
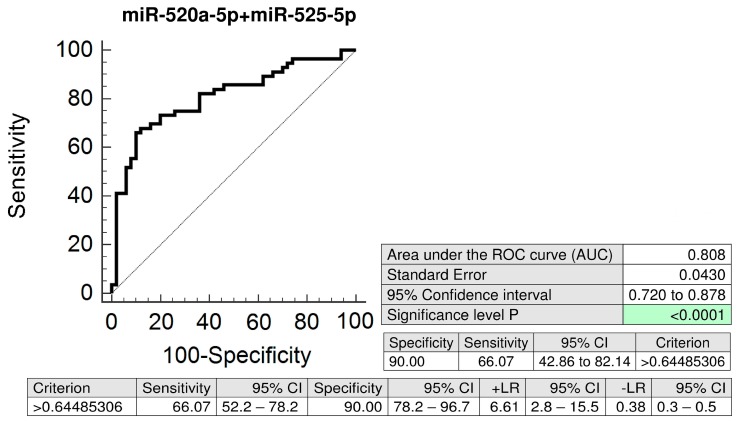
ROC curves—the best combination of C19MC microRNA biomarkers—evaluation of the potential of the first trimester C19MC microRNA screening in plasma exosomes to predict later onset of GH. The combination of miR-520a-5p and miR-525-5p showed the best predictive performance for the prediction of the later occurrence of GH (66.07% sensitivity at 10.0% FPR). GH: gestational hypertension.

**Figure 4 ijms-20-02972-f004:**
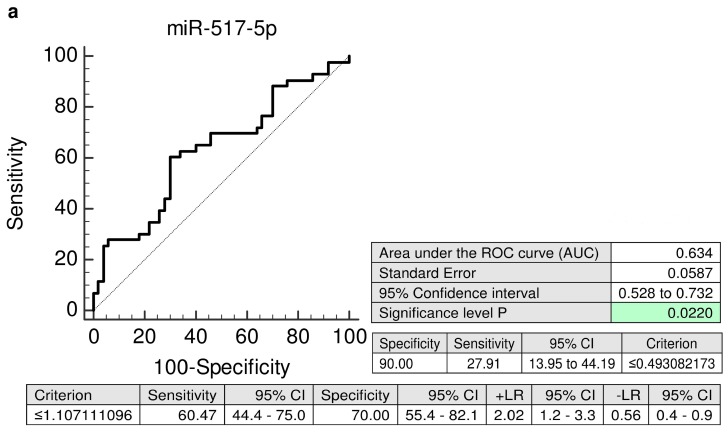
ROC curves—individual C19MC microRNA biomarkers—evaluation of the potential of the first trimester C19MC microRNA screening in plasma exosomes to predict subsequent onset of PE. Decreased levels of miR-517-5p, miR-520a-5p, and miR-525-5p were detected in women destinated to develop PE when the comparison to the controls was performed (**a**–**c**). PE: preeclampsia.

**Figure 5 ijms-20-02972-f005:**
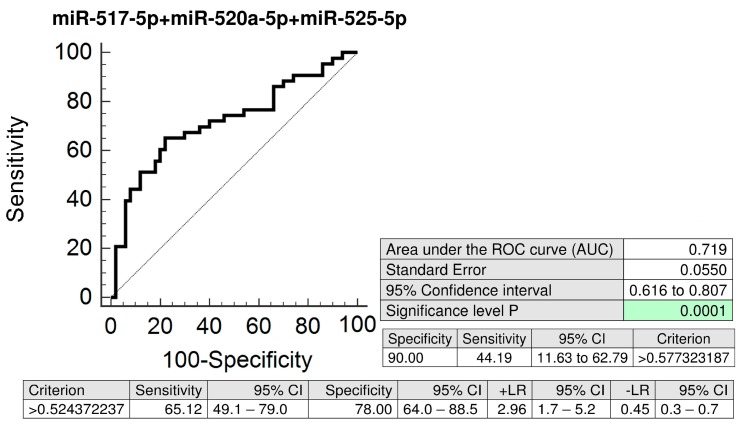
ROC curves—the best combination of C19MC microRNA biomarkers—evaluation of the potential of the first trimester C19MC microRNA screening in plasma exosomes to predict subsequent onset of PE. The combination of miR-517-5p, miR-520a-5p, and miR-525-5p showed the best predictive performance for the prediction of the later occurrence of PE (44.19% sensitivity at 10.0% FPR). PE: preeclampsia.

**Figure 6 ijms-20-02972-f006:**
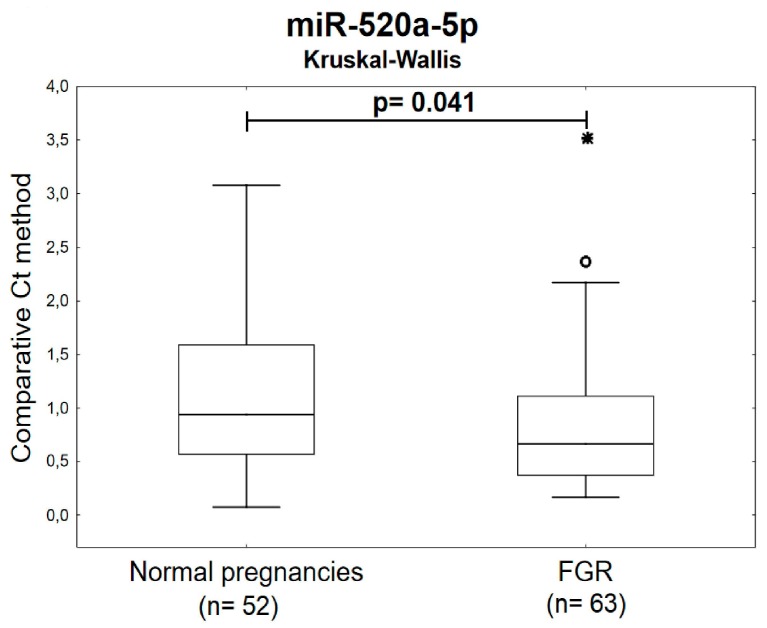
Down-regulation of miR-520a-5p in plasma exosomes during the first trimester of gestation in women with subsequent onset of FGR. Decreased levels of miR-520a-5p were observed in circulating plasma exosomes within 10 to 13 weeks of gestation in women affected with FGR when the comparison to the controls was performed using the Kruskal-Wallis test. FGR: fetal growth restriction. Outliers are marked by circles (⸰), and extremes by asterisks (*)

**Figure 7 ijms-20-02972-f007:**
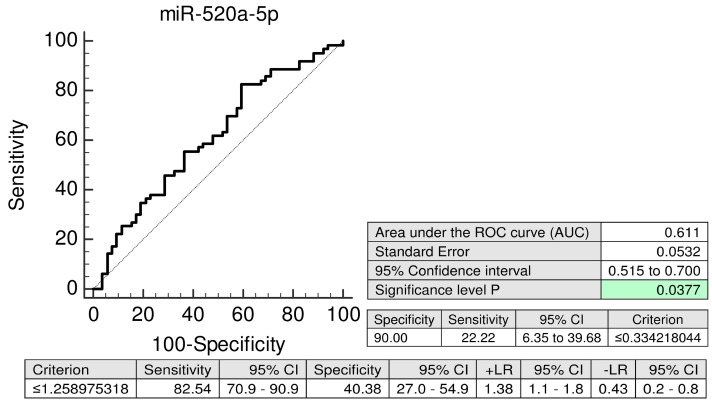
ROC curves—evaluation of the potential of the first trimester miR-520a-5p biomarker screening in plasma exosomes to predict subsequent onset of FGR. Decreased levels of miR-520a-5p were detected in women destinated to develop FGR when the comparison to the controls was performed. FGR: fetal growth restriction.

**Table 1 ijms-20-02972-t001:** Clinical characteristics of the controls and complicated pregnancies.

	Control Group 1 (*n* = 50)	Control Group 2 (*n* = 52)	PE (*n* = 43)	FGR (*n* = 63)	GH (*n* = 57)	*p*-Value
At sampling						
Maternal age (years); mean ± SE	31.88 ± 0.56	31.21 ± 0.56	32.34 ± 0.73	33.42 ± 0.57	32.15 ± 0.63	-
median (range)	32 (23–39)	31 (23–41)	31 (23–46)	33 (22–44)	32 (22–42)	-
Gestational age (weeks); mean ± SE	10.69 ± 0.14	10.40 ± 0.08	10.82 ± 0.18	10.37 ± 0.07	10.84 ± 0.13	-
median (range)	10.29 (9.86–13.71)	10.29 (10.0–13.43)	10.29 (9.86–13.86)	10.29 (9.86–13.29)	10.43 (9.71–14.0)	-
At delivery						
Gestational age (weeks); mean ± SE	40.11 ± 0.11	39.75 ± 0.15	36.0 ± 0.49	36.68 ± 0.30	39.32 ± 0.16	PE vs. Control group1; *p* < 0.001 FGR vs. Control group2; *p* < 0.001
median (range)	40.29 (37.71-42.0)	40.0 (37.29–41.86)	36.71 (28.0–40.71)	37.29 (28.29–40.29)	39.14 (36.0–41.71)	PE vs. Control group1; *p* < 0.001 FGR vs. Control group2; *p* < 0.001
Blood pressure (mmHg)						
Systolic; mean ± SE	122.06 ± 1.58	122.05 ± 1.74	154.2 ± 2.04	124.91 ± 2.27	148.97 ± 2.42	PE vs. Control group1; *p* < 0.001 GH vs. Control group1; *p* < 0.001
median (range)	120 (100–142)	120 (90–148)	150 (133–186)	125 (86–177)	150 (107–200)	PE vs. Control group1; *p* < 0.001 GH vs. Control group1; *p* < 0.001
Diastolic; mean ± SE	76.28 ± 0.94	77.64 ± 1.28	99.74 ± 1.38	79.7 ± 1.56	92.75 ± 1.43	PE vs. Control group1; *p* < 0.001 GH vs. Control group1; *p* < 0.001
median (range)	76 (65–88)	78 (58–93)	100 (80–120)	80 (59–109)	95 (70–114)	PE vs. Control group1; *p* < 0.001 GH vs. Control group1; *p* < 0.001
Fetal birth weight (grams); mean ± SE	3521.02 ± 47.32	3476.42 ± 46.33	2551.90 ± 143.12	2179.46 ± 60.72	3503.87 ± 64.55	PE vs. Control group1; *p* < .001 FGR vs. Control group2; *p* < 0.001
median (range)	3520 (2780–4240)	3440 (2690–4290)	2565 (930–4460)	2260 (746–3230)	3480 (2510–4670)	PE vs. Control group1; *p* < 0.001 FGR vs. Control group2; *p* < 0.001
Mode of delivery						
Vaginal	36 (72.0%)	43 (82.69%)	7 (16.28%)	16 (25.4%)	36 (71.93%)	PE vs. Control group1; *p* < 0.001 FGR vs. Control group2; *p* < 0.001
CS	14 (28.0%)	9 (17.31%)	36 (83.72%)	47 (74.6%)	16 (28.07%)	PE vs. Control group1; *p* < 0.001 FGR vs. Control group2; *p* < 0.001
Fetal sex						
Boy	20 (40.0%)	27 (51.92%)	19 (44.19%)	32 (56.14%)	30 (47.62%)	-
Girl	30 (60.0%)	25 (48.08%)	24 (55.81%)	25 (43.86%)	33 (52.38%)	-
Primiparity						
Yes	21 (42.0%)	31 (59.62%)	33 (76.74%)	35 (61.4%)	38 (60.32%)	-
No	20 (58.0%)	21 (40.38%)	10 (23.26%)	22 (38.6%)	25 (39.68%)	-

Continuous variables, compared using the ANOVA test or the Kruskal-Wallis test, are presented as mean ± SE and median (range), respectively. Categorical variables, presented as number (percent), were compared using Chi-squared test. PE, preeclampsia; GH, gestational hypertension; FGR, fetal growth restriction; CS, Caesarean section; SE, standard error.

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
