# Peer review of "The Prediction of Gestational Hypertension, Preeclampsia and Fetal Growth Restriction via the First Trimester Screening of Plasma Exosomal C19MC microRNAs"

_ijms, 2019, doi:10.3390/ijms20122972_

Round 1

Reviewer 1 Report

Ilona et al attempted to study the prediction of gestational hypertension, preeclampsia and fetal growth restriction via the first-trimester screening of plasma exosomal C19MC microRNA.

It looks a bit surprised to me that the ROC appeared to be so alarmingly perfect in Figure 2 after adjusting for appropriate covariates. However, in Figure 1i, the data points cannot lead to such a perfect ROC curve in Figure 2.

The authors should describe the details of how they performed the adjustment and should accurately describe why the difference is so huge between before and after covariate adjustment.

It is necessary to generate a new independent dataset to see if the perfect performance can be reproducible. In a similar study published in PLoS One by the same group (PLoS One. 2017; 12(2): e0171756.), the performance seemed much worse performance with the use the same markers.

Author Response

Ilona et al attempted to study the prediction of gestational hypertension, preeclampsia and fetal growth restriction via the first-trimester screening of plasma exosomal C19MC microRNA.

It looks a bit surprised to me that the ROC appeared to be so alarmingly perfect in Figure 2 after adjusting for appropriate covariates. However, in Figure 1i, the data points cannot lead to such a perfect ROC curve in Figure 2.

Concerning reviewer´s 2 comment we performed Shapiro-Wilk test, and choose just one test based on the normality of the data. Please see the revised text below.

4.6. Statistical analysis

Normality of the data was assessed using Shapiro-Wilk test, which indicated that our experimental data did not follow a normal distribution. Therefore, C19MC microRNA levels were primarily compared between groups using non-parametric test (the Kruskal-Wallis one-way analysis of variance with post-hoc test for the comparison among multiple groups). The significance level was established at a p-value of p < 0.05.

Table S1. The assessment of the normality of experimental data using Shapiro-Wilk test.

Shapiro-Wilk test

miR-516b-5p

W=0.78548

p<0.001

miR-517-5p

W=0.39129

p<0.001

miR-518b

W=0.41613

p<0.001

miR-520a-5p

W=0.22953

p<0.001

miR-520h

W=0.60976

p<0.001

miR-525-5p

W=0.19550

p<0.001

Normality of the data was assessed using Shapiro-Wilk test, which indicated that our experimental data did not follow a normal distribution.

The normality of the data was also assessed in individual subgroups (NP, GH, PE, and FGR), in which we did not find normal distribution as well.

The authors should describe the details of how they performed the adjustment and should accurately describe why the difference is so huge between before and after covariate adjustment.

Concerning reviewer´s 2 comment we choose just one test based on the normality of the data (the Kruskal-Wallis one-way analysis of variance with post-hoc test for the comparison among multiple groups). ANOVA and ANCOVA analyses were removed from the revised manuscript.

It is necessary to generate a new independent dataset to see if the perfect performance can be reproducible. In a similar study published in PLoS One by the same group (PLoS One. 2017; 12(2): e0171756.), the performance seemed much worse performance with the use the same markers.

We extended conclusion for the recommendation of the reviewer. Please see below.

5. Conclusions

The down-regulation of miR-517-5p, miR-520a-5p, and miR-525-5p was observed in patients with later occurrence of GH and PE. Maternal plasma exosomal profiling of selected C19MC microRNAs also revealed a novel down-regulated biomarker during the first trimester of gestation (miR-520a-5p) for women destinated to develop FGR. First trimester circulating plasma exosomes possess the identical C19MC microRNA expression profile as placental tissues derived from patients with GH, PE and FGR during labour. The predictive accuracy of first trimester C19MC microRNA screening (miR-517-5p, miR-520a-5p, and miR-525-5p) for the diagnosis of GH and PE was significantly higher in case of expression profiling of maternal plasma exosomes compared to expression profiling of whole maternal plasma samples. Consecutive large scale studies are needed to verify the findings resulting from this pilot study. Nevertheless, the performance of that kind of studies will be highly demanding, since ten thousands of first trimester plasma samples have to be collected to get sufficient amount of cases who will subsequently develop pregnancy-related complications such as GH, PE or FGR. For the purpose of this study we collected plasma samples from 4356 women to acquire 163 samples from women that later developed relevant pregnancy-related complications (57 GH, 43 PE, and 63 FGR).

Reviewer 2 Report

This is an interesting manuscript describing the application of exosomal microRNAs as biomarkers for preeclampsia, gestational hypertension, and fetal growth restriction.  

With revision and clarifying the following issues, I suggest this paper be accepted.

introduction

line 29 "during labor" - Please rephrase the sentence here and throughout the article, as the placenta samples were collected after birth and the mode of birth has a significant impact on microRNA signature.

line 50 Wrong word order, correction is needed like this: "...since only paternally inherited alleles are expressed in the placenta due o genomic imprinting" 

results

figures - Please use p<0.05 as well where it's appropriate, as you used that significance level.

line 94 -  "the later onset GH" - This sentence is confusing, write instead: the subsequent onset of GH

discussion

line 202-205 - do you have any explanation for the discrepancy in your data? Did you use different methods/ controls?

line 243 - "FGR risky pregnancies" - wrong expression, please correct:  "miR-520a-5p may be a novel promising placental specific biomarker for FGR with a potential of early stratification of high-risk pregnancies"

methods 

Was the quality of the isolated exosomes checked (by FACS, electron microscopy, etc.)?
statistics  - It is not clear why did you use three kinds of statistical tests. You should have been chosen between the proper parametric / non-parametric tests based on the normality of the data. 

line 247 - Wrong word: it was performed in 6/2011-2/2019 

line 284 - Wrong word order, correction is needed like this: "...were centrifuged twice immediately after collection..."

line 285 - 1200 q - please use rcf/rpm

line 293 - Missing word: spike-in 

line 300 - Missing word: The analyzed C19MC microRNAs... 

line 307 - "stated thermal cycling parameters" - wrong expression, you can use "predefined" for example 

line 338 -  Wrong word: allows creating 

Author Response

Please find the answers to reviewer´s 2 comments in the attachment.

Reviewer 3 Report

This is an interesting study that aimed at verifying whether the quantification of placental specific C19MC microRNAs in plasma exosomes would have diagnostic value in the first trimester for pregnancy-related complications. Maternal plasma was collected from women who later developed GH (n=57), PE (n=43), FGR (n=63), and 102 controls. Exosome profiling was performed with the selection of C19MC microRNAs with diagnostics potential only using real-time RT-PCR. The down-regulation of certain miRNAs was detected in patients who later developed GH, PE and FGR. The authors concluded that the predictive accuracy of first trimester C19MC miRNA screening for the diagnosis of GH and PE is significantly higher in case of expression profiling of maternal plasma exosomes compared to expression profiling of whole maternal plasma samples.

Although the study has many strengths and the results are interesting, the reviewer found several issues that have to be meticulously corrected:

Line 18: There is no analysis of prognostic potential of the investigated miRNAs. Therefore, please delete the term prognostic potential from the whole text.

Line 20: The term case cohort does not exist. Please replace it with relevant term all along the text (e.g. case control study nested in a cohort).

Line 58: The reviewer found other publication on first trimester exosome profiling. Please correct the statement and include relevant reference(s).

Line 72: Please present all the data, also with no statistically significant changes. These latter can be included in a supplementary material.

Figure 1 b and h: Box Cox transformation was not applied before ANOVA as described in the materials and methods.

Figure 1 e and f: The data does not seem to be affected by the Box Cox transformation.

Figure 1 c, f and i: The data does not seem to be affected by the adjustment for covariates.

Figure 2 and 4: ROC curves have dramatic changes after data adjustment to covariates. However, as can be seen from Figure 1, data is not changed after adjustment to covariates. Also, in the table it can be seen that these covariates are essentially the same in the groups. As such, the reviewer does not understand these huge improvements.

Figure 3, 5 and 7: Why unadjusted data was used if adjusted, non-combined data was found to be superior?

Figure 5: Here data adjustment seems to affect data.

Line 202: How authors can provide an explanation for that the data presented herein are just the opposite they published in a previous study?

Line 247. The first two sentences need to be merged.

Lines 253 and 262: Subanalyses are warranted to see in which PE (based on GA) or FGR (based on PI) subtypes are the miRNAs most affected.

Line 273: There is no meaning to have two control groups with the same parameters. In order to reach larger statistical power and to reduce bias and reader confusion, please combine the two control groups into one and please redo all the analyses.

Line 277: Please include the ethics committee approval number.

Line 334: It is stated that the Box Cox transformation was performed prior to the ANCOVA test. However, this was also performed prior to the ANOVA test as shown by the figures, which is incorrect.

Line 388: Please use only the term FGR or IUGR.

The English of the manuscript has to be reviewed by native speaker since in certain parts it has to be improved.

Overall, the reviewer suggests a meticulous major revision for the correction of analysis, data interpretation and presentation.

Author Response

Please find the answers to reviewer´s  3 comments in the attachment.

This is an interesting study that aimed at verifying whether the quantification of placental specific C19MC microRNAs in plasma exosomes would have diagnostic value in the first trimester for pregnancy-related complications. Maternal plasma was collected from women who later developed GH (n=57), PE (n=43), FGR (n=63), and 102 controls. Exosome profiling was performed with the selection of C19MC microRNAs with diagnostics potential only using real-time RT-PCR. The down-regulation of certain miRNAs was detected in patients who later developed GH, PE and FGR. The authors concluded that the predictive accuracy of first trimester C19MC miRNA screening for the diagnosis of GH and PE is significantly higher in case of expression profiling of maternal plasma exosomes compared to expression profiling of whole maternal plasma samples.

Although the study has many strengths and the results are interesting, the reviewer found several issues that have to be meticulously corrected:

Line 18: There is no analysis of prognostic potential of the investigated miRNAs. Therefore, please delete the term prognostic potential from the whole text.

As suggested, it was corrected.

Abstract:

The aim of the study was to verify if quantification of placental specific C19MC microRNAs in plasma exosomes would be able to differentiate during early stages of gestation between patients subsequently developing pregnancy-related complications and women with normal course of gestation and if this differentiation would lead to the improvement of the diagnostical potential.

1. Introduction

 The aims of the current study are to explore A) if quantification of placental specific C19MC microRNAs in plasma exosomes would be able to differentiate during early stages of gestation between patients subsequently developing pregnancy-related complications and women with normal course of gestation and B) if this differentiation would lead to the improvement of their diagnostical potential (better detection rate).

Line 20: The term case cohort does not exist. Please replace it with relevant term all along the text (e.g. case control study nested in a cohort).

 As suggested, it was corrected.

Abstract:

The case control study nested in a cohort involved women that later developed GH (n=57), PE (n=43), FGR (n=63), and 102 controls.

4. Materials and Methods

4.1. Patients cohort

The case control study nested in a cohort involved women that later developed relevant pregnancy-related complications (57 GH, 43 PE, and 63 FGR) [38-41].

Line 58: The reviewer found other publication on first trimester exosome profiling. Please correct the statement and include relevant reference(s).

As suggested, it was corrected.

1. Introduction

To date, little data on first trimester exosome microRNA profiling is available in women with subsequent development of pregnancy-related complications such as gestational hypertension (GH), preeclampsia (PE) and/or fetal growth restriction (FGR) [31,32].

References:

31. Salomon, C., Guanzon, D., Scholz-Romero, K., Longo, S., Correa, P., Illanes, S.E., Rice, G.E. Placental Exosomes as Early Biomarker of Preeclampsia: Potential Role of Exosomal MicroRNAsAcross Gestation. J Clin Endocrinol Metab 2017, 102, 3182-3194. 

 32. Devor, E., Santillan, D., Scroggins, S., Warrier, A., Santillan, M. Trimester-specific plasma exosome microRNA expression profiles in preeclampsia. J Matern Fetal Neonatal Med 2019, 30, 1-9.

Line 72: Please present all the data, also with no statistically significant changes. These latter can be included in a supplementary material.

 As suggested, newly the data with non-significant results are presented in a supplementary material.

Table S2. The presentation of no statistically significant results.

miR-516b-5p

miR-518b

miR-520h

GH vs NP

p= 0.194

p= 1.0

p= 0.741

PE vs NP

p= 1.0

p= 0.735

p= 1.0

miR-516b-5p

miR-517-5p

miR-518b

miR-520h

miR-525-5p

FGR vs NP

p= 0.840

p= 0.937

p= 0.278

p= 0.282

p= 0.633

No difference in microRNA expression was observed in circulating plasma exosomes within 10 to 13 weeks of gestation in women affected with GH, PE or FGR when the comparison to the controls was performed using non-parametric statistical test (the Kruskal-Wallis test). PE, preeclampsia; GH, gestational hypertension; FGR, fetal growth restriction; NP, normal pregnancies.

Figure 1 b and h: Box Cox transformation was not applied before ANOVA as described in the materials and methods. Figure 1 e and f: The data does not seem to be affected by the Box Cox transformation. Figure 1 c, f and i: The data does not seem to be affected by the adjustment for covariates. Figure 3, 5 and 7: Why unadjusted data was used if adjusted, non-combined data was found to be superior? Figure 5: Here data adjustment seems to affect data. Figure 2 and 4: ROC curves have dramatic changes after data adjustment to covariates. However, as can be seen from Figure 1, data is not changed after adjustment to covariates. Also, in the table it can be seen that these covariates are essentially the same in the groups. As such, the reviewer does not understand these huge improvements. Line 334: It is stated that the Box Cox transformation was performed prior to the ANCOVA test. However, this was also performed prior to the ANOVA test as shown by the figures, which is incorrect.

Please find the answer to all of these comments below:

Concerning reviewer´s 2 comment we performed Shapiro-Wilk test, and choose just one test based on the normality of the data. Please see the revised text below.

4.6. Statistical analysis

Normality of the data was assessed using Shapiro-Wilk test, which indicated that our experimental data did not follow a normal distribution. Therefore, C19MC microRNA levels were primarily compared between groups using non-parametric test (the Kruskal-Wallis one-way analysis of variance with post-hoc test for the comparison among multiple groups). The significance level was established at a p-value of p < 0.05.

Table S1. The assessment of the normality of experimental data using Shapiro-Wilk test.

Shapiro-Wilk test

miR-516b-5p

W=0.78548

p<0.001

miR-517-5p

W=0.39129

p<0.001

miR-518b

W=0.41613

p<0.001

miR-520a-5p

W=0.22953

p<0.001

miR-520h

W=0.60976

p<0.001

miR-525-5p

W=0.19550

p<0.001

Normality of the data was assessed using Shapiro-Wilk test, which indicated that our experimental data did not follow a normal distribution.

The normality of the data was also assessed in individual subgroups (NP, GH, PE, and FGR), in which we did not find normal distribution as well.

Line 202: How authors can provide an explanation for that the data presented herein are just the opposite they published in a previous study?

 As suggested, we added the possible explanation for that situation.

3. Discussion

We believe that dissimilar expression profiles of C19MC microRNAs between maternal plasma and maternal plasma exosomes can be influenced by compilations stemming from several factors. At the very least, an expression of particular C19MC microRNA in maternal plasma is represented by the total sum of expression of this particular C19MC microRNA in individual cells located in different areas of placenta, which currently undergo apoptosis, release placental debris into the maternal circulation, and actively secrete exosomes mediating intercellular communication.

Line 247. The first two sentences need to be merged.

As suggested, it was corrected.

The study had a retrospective design, it was performed in 6/2011-2/2019.

Lines 253 and 262: Subanalyses are warranted to see in which PE (based on GA) or FGR (based on PI) subtypes are the miRNAs most affected.

As suggested, we performed statistical subanalyses in the following individual subgroups (please see the table below):

-        13 mild PE vs. 30 severe PE

-        10 early PE vs. 33 late PE

-        PI in arteria umbilicalis: 23 FGR cases (abberant) vs. 40 FGR cases (normal)

-        PI arteria cerebri media (below 5th percentile): 14 FGR cases (abberant) vs. 49 FGR cases (normal)

-        CPR (below 5th percentile): 47 FGR cases (abberant) vs. 16 FGR cases (normal)

Due to low number of individuals in particular subgroups (i. e. 10 early PE, 13 mild PE, etc.) we do not attach to these data a great importance. We would like to suggest to make the decision by the Editorial Office, if these data would be included in a supplementary material or not.

We did not perform statistical subanalyses due to very low numbers of patients in the following individual subgroups:

-        4 early FGR vs. 59 late FGR

-        PI arteria uterine (above 95th percentile): 4 FGR cases (abberant) vs. 59 FGR cases (normal)

-        PI Ductus venosus (>1): 3 FGR cases (abberant) vs. 60 FGR cases (normal)

-        Absent and/or zero diastolic flow in the arteria umbilicalis: 3 FGR cases (abberant) vs. 60 FGR cases (normal)

-        The presence of unilateral/bilateral diastolic notch in the uterine artery: 5 FGR cases (abberant) vs. 58 FGR cases (normal)

-        An absence of flow during atrial contraction in Ductus venosus: 1 FGR case (abberant) vs. 62 FGR cases (normal)

Table S3. C19MC microRNA expression in circulating plasma exosomes within 10 to 13 weeks of gestation with regard to individual subgroups of pregnancy-related complications

miR-516b-5p

miR-517-5p

miR-518b

miR-520a-5p

miR-520h

miR-525-5p

PE

Mild PE (n=13) vs NP (n= 50)

1.0

0.009 ↓

1.0

0.006 ↓

0.588

0.006 ↓

Severe PE (n= 30) vs NP (n= 50)

1.0

0.748

1.0

0.041 ↓

1.0

0.046 ↓

Early PE (n= 10) vs NP (n= 50)

0.448

0.634

0.055 ↑

0.339

1.0

1.0

Late PE (n= 33) vs NP (n= 50)

1.0

0.105

1.0

0.003 ↓

0.316

0.001 ↓

FGR

PI arteria umbilicalis

       Abberant PI (n= 23) vs NP (n= 52)

1.0

1.0

1.0

0.143

1.0

0.336

       Normal PI (n= 40) vs NP (n= 52)

1.0

1.0

1.0

0.271

0.057 ↓

0.211

PI arteria cerebri media

       Abberant PI (n= 14) vs NP (n= 52)

1.0

0.820

1.0

1.0

1.0

1.0

       Normal PI (n= 49) vs NP (n= 52)

0.983

1.0

1.0

0.105

0.304

0.062 ↓

CPR

       Abberant (n= 47) vs NP (n= 52)

1.0

1.0

1.0

0.158

0.295

0.320

       Normal (n= 16) vs NP (n= 52)

0.267

1.0

1.0

0.284

0.429

0.081 ↓

PE, preeclampsia; FGR, fetal growth restriction; NP, normal pregnancies; PI, pulsatility index; CPR, cerebro-placental ratio; ↓, Decreased levels of microRNA in a particular subgroup of pregnancy-related complications when the comparison to the controls was performed using the Kruskal-Wallis test; ↑, Increased levels of microRNA in a particular subgroup of pregnancy-related complications when the comparison to the controls was performed using the Kruskal-Wallis test.

Line 273: There is no meaning to have two control groups with the same parameters. In order to reach larger statistical power and to reduce bias and reader confusion, please combine the two control groups into one and please redo all the analyses.

Unfortunately, the reviewer´s suggested approach is not possible. Please see the explanation in the original manuscript.

4.5. Quantification of plasma exosomal C19MC microRNAs by real-time PCR

 The expression of particular C19MC microRNA in maternal plasma exosomes was determined using the comparative Ct method [54] relative to the expression in the reference sample. RNA isolated from the pool of randomly selected maternal plasma samples derived from women at the first trimester with normal course of gestation was used as a reference sample for relative quantification. Two reference samples were used throughout the study (reference 1: the pool of 5 maternal plasma samples, reference 2: the pool of 8 maternal plasma samples).

Line 277: Please include the ethics committee approval number.

The study was approved by the Ethics Committee of the Third Faculty of Medicine, Prague, Czech Republic (Implication of placenta-specific microRNAs in maternal circulation for diagnosis and prediction of placental insufficiency, date of approval: 7.4.2011). The Approval of the Ethics Committee of the Third Faculty of Medicine was submitted to the Editorial office of IJMS. Unfortunately, it has no approval number.

Line 388: Please use only the term FGR or IUGR.

FGR is currently used term. We corrected one sentence where IUGR instead of FGR was used.

1. Introduction

This study is a follow-up of our previous studies dedicated to first trimester screening of circulating C19MC microRNAs in whole maternal plasma and its potential to predict subsequent onset of gestational hypertension, preeclampsia and/or FGR [6, 9].

Abbreviations

FGR

Fetal growth   restriction

Round 2

Reviewer 3 Report

The authors have revised the manuscript and now I suggest it to be accepted.